# Ten Years of KPC-Kp Bloodstream Infections Experience: Impact of Early Appropriate Empirical Therapy on Mortality

**DOI:** 10.3390/biomedicines10123268

**Published:** 2022-12-16

**Authors:** Silvia Corcione, Ilaria De Benedetto, Nour Shbaklo, Fabio Ranzani, Simone Mornese Pinna, Anna Castiglione, Silvia Scabini, Gabriele Bianco, Rossana Cavallo, Stefano Mirabella, Renato Romagnoli, Francesco Giuseppe De Rosa

**Affiliations:** 1Department of Medical Sciences, Infectious Diseases, University of Turin, 10124 Turin, Italy; 2Department of Infectious Diseases, Tufts University School of Medicine, Boston, MA 02111, USA; 3Unit of Clinical Epidemiology, CPO Piemonte, “Città della Salute e della Scienza”, Hospital of Turin, 10131 Turin, Italy; 4Unit of Infectious Diseases, A.O.U. Città della Salute e della Scienza, 10126 Turin, Italy; 5Microbiology and Virology Unit, Azienda Ospedaliero Universitaria Città della Salute e della Scienza di Torino, 10126 Turin, Italy; 6General Surgery 2U and Liver Transplant Unit, Molinette Hospital, A.O.U. Città della Salute e della Scienza, University of Turin, 10124 Turin, Italy; 7Unit of Infectious Diseases, Cardinal Massaia, 14100 Asti, Italy

**Keywords:** bloodstream infections, resistance, antibiotic therapy, KPC

## Abstract

Background. In K. pneumoniae KPC (KPC-Kp) bloodstream infections (BSI), INCREMENT CPE score >7, Charlson Comorbidity Index (CCI) ≥3 and septic shock are recognized predictors of mortality, with a possible beneficial effect of combination therapy in seriously ill patients. Materials and Methods. We conducted a ten-year retrospective study including all KPC-Kp BSI in patients ≥18 years of age with the aim to evaluate the characteristics and impact of appropriate empirical therapy, either monotherapy or combination therapy, and targeted therapy on mortality. Appropriate therapy was defined as at least one active antimicrobial agent with in vitro activity against KPC-kp demonstrated by susceptibility testing, administered within 48 h from blood culture collection. Results. The median age of the 435 analyzed patients was 66.09 years (IQR 54.87–73.98). The median CCI was 4. KPC-Kp colonization was present in 324 patients (74.48%). The probable origin of the KPC-Kp BSI was not identified in 136 patients (31.26%), whereas in 120 (27.59%) patients, it was CVC-related, and in 118 (27.13%), it was respiratory. Source control was achieved in 87 patients (72.5%) with CVC-related KPC-Kp BSI. The twenty-eight-day survival was 70.45% for empirical monotherapy, 63.88% for empirical combination therapy and 57.05% for targeted therapy (*p* = 0.0399). A probable source of KPC-Kp BSI other than urinary, CVC or abdominal [aHR 1.64 (IC 1.15–2.34) *p* = 0.006] and deferred targeted therapy [HR 1.67 (IC 1.12–2.51), *p*= 0.013] emerged as predictors of mortality, whereas source control [HR 0.62 (IC 0.44–0.86), *p* = 0.005] and ceftazidime/avibactam administration in empirical therapy [aHR 0.37 (IC 0.20–0.68) *p* = 0.002] appeared as protective factors. Discussion. These data underline the importance of source control together with timing appropriateness in the early start of empirical therapy over the choice of monotherapy or combination therapy and the use of ceftazidime/avibactam against KPC-Kp BSI.

## 1. Introduction

Since the first occurrence of *Klebsiella pneumoniae* carbapenemase (KPC-Kp)-producing *Klebsiella pneumoniae* in North Carolina, USA, in 1996, KPC-Kp has spread worldwide, with the first outbreaks reported in 2005 [1,2,3,4]. In 2020, according to European Center for Disease Prevention and Control (ECDC) Surveillance [5], the incidence of endemic *K. pneumoniae* carbapenem-resistant isolates increased up to 66.3% in Greece, 29.5% in Italy, 48.3% in Romania, 28.1% in Bulgaria. KPC-Kp infections are a major threat among patients admitted to acute care and long-term hospitals, with a mortality rate ranging from 20% to 70% [6,7,8,9,10]. It is known that gut colonization precedes KPC-kp bloodstream infection (BSI). Specific clinical scores have been determined to define high-risk subgroups with an increased risk for KPC-Kp-related infections or mortality. In a high endemic setting, the Pitt bacteremia score [11] and the INCREMENT-CPE score (ICS) have been validated to predict mortality in patients with carbapenemase-producing *Enterobacterales* BSI [12]. The ICS has also been proposed as a practical tool to orient the choice between monotherapy and combination therapy according to the mortality risk group, reserving the combination therapy to high-ICS risk groups [13].

In fact, several retrospective studies mostly including KPC-Kp BSI and other types of infections suggested that combination therapy reduced the mortality compared to monotherapy [13]. Predictors of mortality were septic shock, inappropriate empirical therapy, Charlson comorbidity index ≥3, high APACHE III and SAPS II scores, neutropenia, mechanical ventilation, corticosteroid administration and parenteral nutrition, whereas protective factors were targeted combination therapy including tigecycline, colistin and meropenem [7,14,15,16,17]. On the other hand, in a cohort of 437 patients with CRE-BSI, a lower mortality rate was observed for appropriate (defined as therapy administered within 5 days from the bloodstream infection onset) versus inappropriate therapy. As only retrospective studies have been provided concerning this matter, the roles of combination treatment and monotherapy for KPC-Kp infections are still a matter of debate. Data from multicenter studies describe an advantage in terms of survival for patients treated with ceftazidime/avibactam alone or in any combination therapy [18,19]. Therefore, with this retrospective study, we aimed to investigate the outcome and characteristics of patients with documented BSI due to KPC-Kp who were administered different antibiotic regimens during a single-center 10-year experience.

## 2. Materials and Methods

### 2.1. Design of the Study

We conducted a retrospective study at AOU Città della Salute e della Scienza, Torino, Italy, including all nosocomial cases of KPC-Kp BSI in patients ≥18 years of age between January 2010 and December 2019.

The primary objective was to evaluate the impact of appropriate empirical therapy, either monotherapy or combination therapy, on mortality and to evaluate survival on day 7, day 14 and day 28 from BSI onset in patients with nosocomial KPC-Kp BSI.

The secondary objectives were:to evaluate the impact of ceftazidime/avibactam in empirical therapy, either monotherapy or combination therapy, on mortality.to evaluate the impact of nephrotoxicity on mortality in a subgroup of patients treated with colistin or aminoglycoside as a part of combination regimens.

### 2.2. Inclusion Criteria and Definitions

Inclusion criteria were: all KPC-Kp nosocomial BSI defined as at least two blood cultures positive for KPC-producing *Klebsiella pneumoniae* collected after ≥2 days from admission and concomitant Systemic Inflammatory Response Syndrome (SIRS) signs [corporeal temperature <36 °C or >38 °C, pulse rate >90 beats/minute, respiratory rate >20 breaths/minute, blood cells count <4000 cells/mm^3^ or >12,000 cells/mm^3^ or >10% immature neutrophils (band form)]. Empirical therapy was considered as any antibiotic treatment started before obtaining the blood cultures results and administered for at least 48 h after blood culture collection. Combination therapy was defined as the administration of at least two antibiotics for ≥48 h. Appropriate therapy was defined as the administration of at least one active in vitro antibiotic for ≤48 h from blood culture collection. Nephrotoxicity was considered according to the definition by the modified KDIGO guidelines as a 1.5-fold increase in baseline serum creatinine levels within 7 days after treatment initiation. Source control was considered effective when it was achieved in the first 48 h after blood culture collection.

Targeted therapy was defined as the administration of any active antibiotic against KPC-Kp once the blood culture results were available to the clinicians.

### 2.3. Data Collection

We collected demographical and clinical data at KPC-Kp BSI onset, including Charlson comorbidity index and comorbidities. KPC-Kp surveillance of other than stool sites was recorded, including urinary, respiratory and abdominal drainage carriage, surgical/medical/ICU admission, length of stay, source of KPC-Kp BSI. The presence of a central venous catheter (CVC), mechanical ventilation, hemodialysis, continuous veno-venous hemofiltration (CVVH), extra-corporeal membrane oxygenation (ECMO) was recorded. Data on antimicrobials used previously as part of a combination treatment for KPC-kp BSI (Meropenem, Imipenem, Ertapenem, Tigecycline, Colistin, Amikacin, Gentamicin, Piperacillin/Tazobactam, Trimethoprim/Sulfamethoxazole, Fosfomycin and Ceftazidime/Avibactam), doses, timing, and appropriateness of the empirical and targeted therapies, both in monotherapy and in combination therapy, were collected as well. The study was approved by the Intercompany Ethics Committee on 13 March 2020, protocol number 0027840.

### 2.4. Microbiological Methods

The blood cultures were incubated with a BactAlert Virtuo instrument (Marcy l’Ètoile, France) according to European Committee on Antimicrobial Susceptibility Testing (EUCAST) breakpoint tables. The identification of the microorganism was conducted with MALDI-TOF MS analysis (Matrix-Assisted Laser Desorption Ionization-Time-of-Flight MS). Bruker Microflex LT mass spectrometer (Bruker Daltonik, Bremen, Germany). FlexControl 3.3 and Maldi Biotyper 3.0 software (Bruker Daltonik) were applied to acquire the spectra and for the identification of the isolates.

The presence of resistance-associated genes was identified through qualitative polymerase chain reaction (Xpert Carba-R^®^-Chepeid) to identify carbapenemases (KPC, NDM, VIM, IMP-1 and OXA-48). 

### 2.5. Statistical Analysis

The demographic and clinical characteristics of patients were summarized through absolute frequencies and percentages for the qualitative variables and through percentiles (median, first quartile-third quartile) for the quantitative variables. Overall survival was defined as the time interval between the KPC-Kp BSI onset and the date of death or the discharge date. The survival functions were estimated by the Kaplan–Meier method and compared with the log rank test. The crude and adjusted effects of different prognostic factors on mortality were estimated using Cox models. Statistical analysis was run using SAS version 9.4 and SPSS version 28 [20,21].

## 3. Results

### 3.1. Baseline Characteristics of the Overall Population

We identified 453 patients with KPC-kp BSI. Of them, 18 patients were excluded because of the lack of essential information. We included in the analysis 435 patients (323 patients treated with empirical therapy, of which 171 received appropriate empirical therapy, and 112 patients treated with targeted treatment; Table 1).

Among the patients treated with empirical therapy, 127 received empirical monotherapy (29.2%), of which 36 received appropriate therapy (8.28%), whereas 196 (45.06%) patients received empirical combination therapy, of which 165 received appropriate therapy (37.93%). Data on significant baseline characteristics are shown in Table 1.

Overall, 273 patients (62.76%) were male, and their median age was 66.09 years (IQR 54.87–73.98). Of them, 146 patients (33.56%) were admitted to the medical ward, 166 patients (37.24%) to a surgical ward, and 127 (29.2%) to the intensive care unit (ICU). The median length of stay was 46 days (IQR 27–76), and median the time to KPC-Kp BSI onset from admission was 19 days (IQR 11–37). The median Charlson comorbidity index was 4. Chronic kidney disease was present in 108 patients (24.83%); 108 patients (24.83%) had congestive heart failure, 102 (23.45%) had chronic lung disease, 97 (22.30%) had diabetes mellitus, 55 (12.64%) were solid-organ transplant recipient, 44 (10.11%) had liver disease, 44 (10.11%) had hematological malignancies, 27 (6.21%) had metastatic malignancies, 20 (4.6%) had severe neutropenia, 2 (0.46%) had HIV infection. At the time of the BSI onset, a KPC-Kp rectal swab was positive in 324 patients (74.48%). The median time to KPC-Kp BSI onset from hospital admission was 21 days (IQR 12–37) in colonized patients and 17 days (IQR 8–38) in not-colonized patients.

The CVC was considered a probable source of KPC-Kp BSI in 120 patients (27.59%), the respiratory tract in 118 patients (27.13%), the abdomen in 58 patients (13.33%), the urinary tract in 40 patients (9.2%), whereas in 136 patients (31.26%) the source of BSI was not identified. Source control was achieved in 125 patients (28.74%), most likely in patients with CVC-related KPC-Kp BSI (72.5%) followed by intra-abdominal infections (20.69%).

Among patients treated with empirical or targeted treatment, a statistically significant difference was observed in patients with diabetes mellitus (*p* = 0.041), KPC-Kp colonization (*p* = 0.004), KPC from a respiratory (*p* = 0.001) or abdominal source (*p* = 0.012), and urinary catheterization (*p* = 0.001; Table 1). We observed that 41.67% of the patients admitted to medical wards received an appropriate empirical monotherapy compared to 25% of the patients admitted to surgical wards and 33.33% of those in ICU. Even in the combination group, an appropriate treatment was observed in 29.09% of the patients in the medical wards compared to 36% in surgical wards and 33.94% in ICU.

### 3.2. Mortality

#### 3.2.1. Impact of Appropriate Empirical Therapy, Either Monotherapy or Combination Therapy, on Mortality in Patients with Nosocomial-Onset KPC-Kp BSI

The overall mortality during hospital stay was 41.84% (182 patients). Among patients who received an appropriate empiric treatment, the 7-day survival was 91.7% for patients receiving monotherapy and 91.5% for those administered combination therapy. After 14 days, the survival was 85.8% for patients receiving empirical appropriate monotherapy and 78.8% for patients receiving empirical appropriate combination therapy; after 28 days, the survival was 70% for patients receiving monotherapy and 65.1% for those receiving combination therapy (Figure 1). The difference in survival between patients receiving appropriate empirical monotherapy and those receiving appropriate empirical combination therapy was not statistically significant (*p* = 0.7694) (Table 2).

The difference in survival among the empirical monotherapy, empirical combination therapy and targeted therapy groups was statistically significant, in favor of patients administered appropriate empirical therapy (*p* = 0.0399). (Table 3). At 28 days, the survival was 70.45% for empirical patients receiving monotherapy, 63.88% for those receiving empirical combination therapy and 57.05% for those receiving targeted therapy (Figure 2).

#### 3.2.2. Crude and Adjusted Effects of Predictors of Mortality

In the univariate model analysis, a probable source of KPC-Kp BSI other than urinary, CVC or abdominal [HR 1.78 (IC 1.31- 2.42), *p* = 0.000] and targeted therapy compared to empirical monotherapy or combination therapy [HR 1.67 (IC 1.12–2.51), *p* = 0.013] emerged as predictors of mortality, whereas source control [HR 0.62 (IC 0.44–0.86), *p* = 0.005] and ceftazidime/avibactam administration in empirical therapy [HR 0.4 (IC 0.22–0.74), *p* = 0.004] appeared as significant protective factors. Charlson comorbidity index ≥4, ICU admission, age, KPC-Kp colonization did not show a significant association with mortality.

Regarding the antibiotic treatment, the use of empirical monotherapy was not associated with a higher mortality compared to that of empirical combination therapy [HR 1.3 (0.90–1.88) *p* = 0.164]. In the multivariate model analysis, a probable source of KPC-Kp BSI other than urinary, CVC or abdominal emerged as an independent predictor of mortality [aHR 1.64 (IC 1.15–2.34) *p* = 0.006] (Table 4),

In the multivariate model analysis, ceftazidime/avibactam administration emerged as an independent protective factor for mortality in overall empirical therapy [aHR 0.37 (IC 0.20–0.68) *p* = 0.002] and in empirical combination therapy [aHR 0.36 (IC 0.17–0.76) *p* = 0.007] (Table 5).

#### 3.2.3. Survival from Nephrotoxicity in Colistin- or Aminoglycoside-Treated Subgroup Patients

In the subgroup analysis, 194 patients were treated with a regimen including colistin or aminoglycoside. Of them, 64 patients (32.99%) developed nephrotoxicity, and 39 patients (60.94%) died during hospitalization. According to the the Kaplan–Meier estimate, the 7-day survival was 90.39% for patients who did not develop nephrotoxicity and 71.84% for patients who developed nephrotoxicity. At the 14-day endpoint, survival was 83.11% for patients who did not develop nephrotoxicity and 62.03% for those who developed nephrotoxicity. At 28 days, survival was 78.53% for those who did not developed nephrotoxicity and 53.30% for those who develop nephrotoxicity. At the 28-day endpoint, survival was 69.34% for patients who did not develop nephrotoxicity and 45.65% for those who developed nephrotoxicity. The difference in survival between patients who developed nephrotoxicity and those who did not develop nephrotoxicity was statistically significant (*p* = 0.0021) (Figure 3, Table 6).

## 4. Discussion

KPC-Kp infections are a major threat among patients admitted to acute care and long-term hospitals, with a mortality rate ranging from 20% to 70%. [6,7,8,9,10]. Usually, KPC-Kp BSI occur in hospitalized patients with several comorbidities, as also confirmed in our setting in which 65% of all KPC-Kp BSI were observed in patients ≥60 years old, and the distribution among medical, surgical and ICU ward was equally divided.

KPC-Kp colonization has been recognized as a predictor of KPC-Kp infections; nonetheless, it does not seem to be a predictor of mortality itself [12,22]. Even in our cohort, 74.48% of the patients were colonized by KPC-KP, but mortality did not differ between the groups. An explanation for this result could be the early administration of appropriate empirical therapy to colonized patients.

We observed a significant impact of the appropriate empirical antimicrobial treatment on survival. Of note, our data highlighted that empirical appropriate monotherapy was non inferior to empirical appropriate combination therapy in terms of survival. This was probably due to the fact that severe infections are time-dependent, recalling the concept “hit hard, early but appropriately” [23]. Such an effect could also be related, in our center, to the high susceptibility retained by KPC-KP to tigecycline and aminoglycosides, mostly used in regimens active against KPC-KP BSI. Moreover, we rarely used colistin monotherapy, sparing nephrotoxicity, which was significantly associated with mortality. Notably, a limitation of this finding could be related to the fact that clinicians might have preferred monotherapy in more clinically stable patients, reserving combination therapy to these more critically ill, as suggested by some studies [14,17,22].

Confirming the existing data related to the Pitt bacteremia score and the INCREMENT CPE score [11,12], in our study, the probable source of KPC-Kp BSI other than urinary, CVC or abdominal emerged as an independent predictor of mortality at univariate analysis. In addition, targeted therapy was associated with increased mortality, whereas the early achievement of source control was associated with a protective effect, as already demonstrated in CVC-related bloodstream infections [24,25,26]. The univariate analysis confirmed the administration of ceftazidime/avibactam as empiric therapy as a protector of mortality. These data were confirmed by a multivariate analysis, in which ceftazidime/avibactam administration emerged as an independent protective factor for mortality in overall empirical therapy [aHR 0.37 (IC 0.20–0.68) *p* = 0.002] and in empirical combination therapy [aHR 0.36 (IC 0.17–0.76) *p* = 0.007], as supported by Tumbarello et al. [20]. A limitation of our study is that we were not able to provide a clinical severity assessment through scores such as Sequential Organ Failure Assessment (SOFA), Acute Physiologic Assessment and Chronic Health Evaluation (APACHE II) or Pitt Bacteremia Score at KPC-Kp BSI presentation, because of the ten-year retrospective nature of the study and the possible incomplete information on clinical records.

## 5. Conclusions

In conclusion, we reported here ten years of experience in the management of KPC-Kp infections. New drugs are available now for the treatment of these infections, which have significantly contributed to improving patient outcome in this setting. Moreover, the importance of source control together with timing appropriateness in the early start of empirical therapy over the choice of monotherapy or combination therapy and the use of new drugs such as ceftazidime/avibactam against *Klebsiella pneumoniae* KPC-bloodstream infections seem to be the basis for an appropriate management of these infections, as suggested by the current guidelines of ESCMID and IDSA [27,28].

## Figures and Tables

**Figure 1 biomedicines-10-03268-f001:**
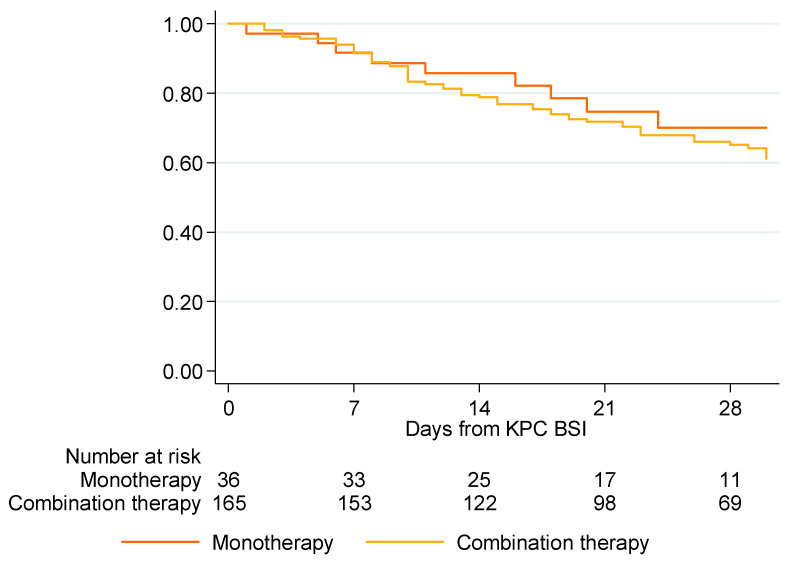
Kaplan–Meier estimate of patient survival for patients receiving appropriate empirical monotherapy and appropriate empirical combination therapy (*n* = 201).

**Figure 2 biomedicines-10-03268-f002:**
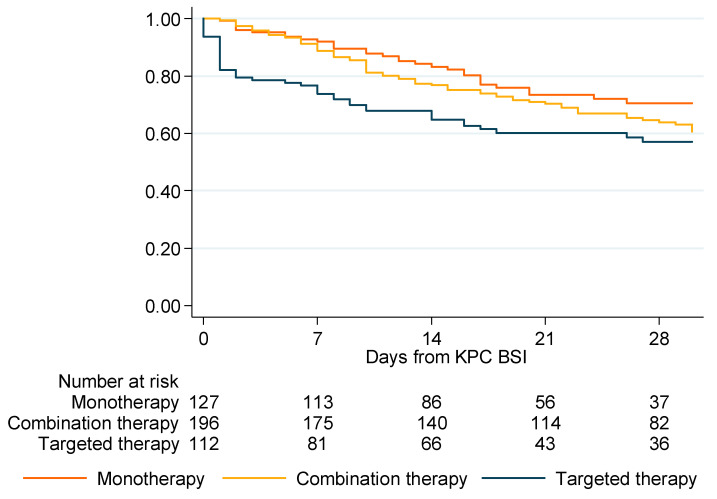
Kaplan–Meier estimate of survival for patients administered empirical monotherapy, empirical combination therapy and targeted therapy (*n* = 435).

**Figure 3 biomedicines-10-03268-f003:**
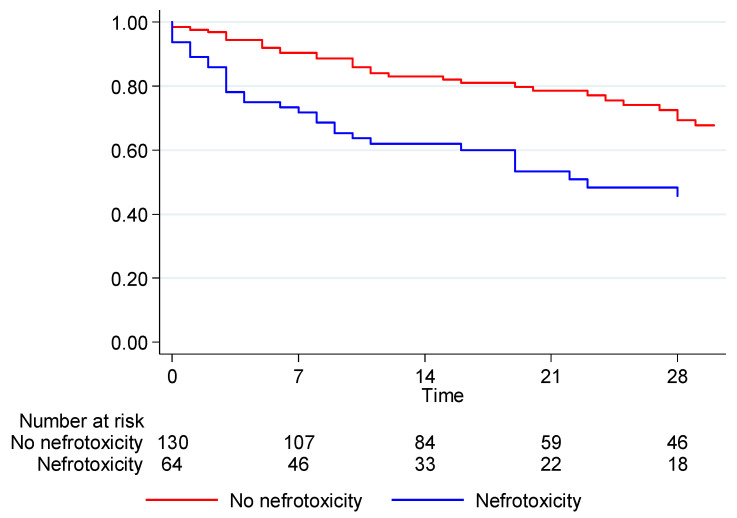
Kaplan–Meier estimate of survival from nephrotoxicity in colistin/aminoglycoside (AG)-treated patients (*n* = 194).

**Table 1 biomedicines-10-03268-t001:** Baseline characteristics of the study population treated with appropriate therapy (*n* = 435).

Variable	Appropriate Empirical Monotherapy (*n* = 36)		Appropriate Empirical Combination Therapy (*n* = 165)		Targeted Therapy (N = 112)		Total (*n* = 435)		*p*-Value
	*n*	%	*n*	%	*n*	%	*n*	%	
**Sex**									
Male	24	66.67	102	61.82	68	60.71	273	62.76	0.92
Female	12	33.33	63	38.18	44	39.29	162	37.24	
**Age**									
<50	7	19.44	34	20.61	16	14.29	74	17.01	0.33
50–59	5	13.89	35	21.21	27	24.11	82	18.85	
60–69	8	22.22	45	27.27	25	22.32	114	26.21	
70–79	13	36.11	40	24.24	32	28.57	126	28.97	
80+	3	8.33	11	6.67	12	10.71	39	8.97	
**Ward of admission**									
Medical ward	15	41.67	48	29.09	28	25	146	33.56	0.009
Surgical ward	9	25	61	36.97	52	46.43	162	37.24	
Intensive care unit	12	33.33	56	33.94	32	28.57	127	29.2	
**Charlson Comorbility index**									
0	3	8.33	8	4.85	3	2.68	21	4.83	
1	4	11.11	24	14.55	15	13.39	48	11.03	
2	6	16.67	28	16.97	20	17.86	67	15.4	
3	5	13.89	30	18.18	21	18.75	87	20	
≥4	18	50	75	45.45	53	47.32	212	48.74	
**KPC-Kp colonization**	25	69.44	138	83.64	78	69.64	324	74.48	0.004
**Probable source of KPC-Kp BSI:**									
**respiratory**	10	27.78	61	36.97	28	25	118	27.13	0.001
**urinary catheter**	29	80.56	142	86.06	100	89.29	362	83.22	0.001
**abdominal**	4	11.11	15	9.09	12	10.61	58	13.33	0.012

**Table 2 biomedicines-10-03268-t002:** Log-rank test for equality of survivor functions, *p*-value = 0.7694.

	Appropriate Empirical Monotherapy	Appropriate Empirical Combination Therapy
**Days**	**OS**	**OS**
**0**	1	1
**7**	0.9167	0.9148
**14**	0.8575	0.7881
**21**	0.7468	0.7186
**28**	0.7001	0.6511

**Table 3 biomedicines-10-03268-t003:** Log-rank test for equality of survivor functions, *p*-value = 0.0399.

	Empirical Monotherapy	Empirical Combination Therapy	Targeted Therapy
**Days**	**OS**	**OS**	**OS**
**0**	1	1	1
**7**	0.9203	0.887	0.7381
**14**	0.8324	0.7684	0.6482
**21**	0.7347	0.703	0.6022
**28**	0.7045	0.6388	0.5705

**Table 4 biomedicines-10-03268-t004:** Crude and adjusted effects of the predictors of mortality (*n* = 435).

	Crude Effect			Adjusted Effect		
	HR	95% CI	*p*	aHR	95% CI	*p*
**Female sex**	0.92	[0.68–1.24]	0.572	1.01	[0.75–1.38]	0.929
**ICU admission**	1.02	[0.75–1.39]	0.892	0.96	[0.70–1.31]	0.799
**Age (every 10 years)**	1.05	[0.95–1.17]	0.322	1.04	[0.93–1.18]	0.473
**Charlson comorbidity index**						
0	1			1		
1	0.81	[0.36–1.81]	0.612	0.69	[0.30–1.54]	0.362
2	0.91	[0.43–1.94]	0.814	0.9	[0.43–1.89]	0.785
3	1.11	[0.54–2.29]	0.781			
≥4	1.14	[0.57–2.25]	0.716			
**Probable source of KPC-Kp BSI other** **than urinary, CVC or abdominal**	1.78	[1.31–2.42]	**0.000**	1.64	[1.15–2.34]	**0.006**
**Source control**	0.62	[0.44–0.86]	**0.005**	0.77	[0.52–1.13]	0.179
**KPC-Kp colonization**	0.91	[0.65–1.29]	0.611	0.98	[0.69–1.39]	0.914
**Appropriate empirical therapy**						
Yes	1					
No	0.8	[0.55–1.15]	0.226			
Only targeted therapy	1.31	[0.93–1.82]	0.119			
**Appropriate empirical therapy**	0.97	[0.73–1.30]	0.855	0.94	[0.70–1.26]	0.684
**Empirical monotherapy**						
Yes	1					
No	1.3	[0.90–1.88]	0.164			
Only targeted therapy	1.67	[1.12–2.51]	**0.013**			
**Type of therapy**						
Appropriate empirical monotherapy	1					
Appropriate empirical combination therapy	1.09	[0.62–1.93]	0.765			
Targeted therapy	1.4	[0.78–2.53]	0.258			
**Ceftazidime/avibactam in empirical therapy**	0.4	[0.22–0.74]	**0.004**			

**Table 5 biomedicines-10-03268-t005:** Adjusted effect of ceftazidime/avibactam administration as a predictor of mortality in overall empirical therapy (*n*= 435), empirical monotherapy (*n* = 127) and empirical combination therapy (*n* = 196).

	Adjusted Effect								
	Overall (*n* = 435)			Empirical Monotherapy (*n* = 127)			Empirical Combination Therapy (*n* = 196)		
	**aHR**	**95% CI**	** *p* **	**aHR**	**95% CI**	** *p* **	**aHR**	**95% CI**	** *p* **
**Female**	1.01	0.74–1.37	0.969	2.61	1.30–5.25	**0.007**	0.68	0.43–1.09	0.107
**Age (every 10 years)**	1.04	0.93–1.18	0.484	1.32	0.99–1.76	0.063	0.95	0.81–1.12	0.558
**ICU admission**	0.93	0.68–1.27	0.648	1.39	0.69–2.79	0.36	0.76	0.49–1.20	0.237
**Charlson comorbidity index**									
0				1.00			1.00		
1	0.65	0.28–1.46	0.294	0.15	0.01–1.47	0.103	1.55	0.41–5.86	0.52
≥2	0.86	0.41–1.82	0.701	0.24	0.07–0.88	0.031	2.08	0.57–7.52	0.266
**Probable source of KPC-Kp BSI other** **than urinary, CVC or abdominal**	1.65	1.17–2.33	**0.005**	1.82	0.91–3.62	0.09	1.64	0.95–2.83	0.077
**Source control**	0.72	0.49–1.06	0.099	1.01	0.48–2.11	0.981	0.63	0.35–1.11	0.112
**KPC-Kp colonization**	1.10	0.77–1.57	0.598	1.47	0.70–3.06	0.308	1.14	0.62–2.10	0.676
**Ceftazidime/avibactam in empirical therapy**	0.37	0.20–0.68	**0.002**	0.26	**0.07–1.02**	**0.054**	0.36	0.17–0.76	**0.007**

**Table 6 biomedicines-10-03268-t006:** Log-rank test for equality of survivor functions, *p*-value= 0.0021.

	No Nephrotoxicity	Nephrotoxicity
**Days**	**OS**	**OS**
**0**	1	1
**7**	0.9039	0.7184
**14**	0.8311	0.6203
**21**	0.7853	0.5330
**28**	0.6934	0.4565

## Data Availability

Partial data were shared with Tumbarello M. for multi-center retrospective studies.

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
