# Peer review of "Ten Years of KPC-Kp Bloodstream Infections Experience: Impact of Early Appropriate Empirical Therapy on Mortality"

_biomedicines, 2022, doi:10.3390/biomedicines10123268_

Round 1

Reviewer 1 Report

The study is well designed and methodology sounds good.

Please add in methods how the statistical analysis was performed (software and version)

Please extend the discussion and add more and different citations (please avoid to add further auto-citations or studies from the same author. It's not forbidden but not classy too).

Please provide the last consultation date for the web url in citation n. 5

Please add the conclusion paragraph.

Author Response

Dear Reviewer, 

We appreciate your time to review our manuscript. Thank you for the critical comments to improve the manuscript. 

Please add in methods how the statistical analysis was performed (software and version)

Thank you for the comment. The softwares used were added along with the version.

Please extend the discussion and add more and different citations (please avoid to add further auto-citations or studies from the same author. It's not forbidden but not classy too).

Thank you for the comment. The citations that some of the authors are co-authors in, are multi-central studies that we had took part of.

Aside from the co-authorship, we believe that they are worth citing since they are robust references of the topic in general and also the region, that enrich the criticism of the discussion.

Please provide the last consultation date for the web url in citation n. 5

Added to bibliography.

Please add the conclusion paragraph.

Added. Thank you.

We hope that we have addressed the comments well. 
Best regard, 

Reviewer 2 Report

The topics is really hot. 

I recommend a few modification.

Double check the English. I recommend that a native speaker of English review the manuscript to improve word choice, sentence structure, and grammar. 

Please describe the abbreviations when you use them for the first time

In the discussion part you need to compare with other studies from your country or surrounding regions. 

I  think that you need three short clear conclusions! Make conclusion which can be used in current clinical use!

Table 5 is hard to read! maximize writing!

Recheck  the references!

Thank you again for the opportunity to review this interesting manuscript. 

Author Response

Dear Reviewer, 

Thank you for the chance to review our manuscript. We appreciate your critical comments. 

Double check the English. I recommend that a native speaker of English review the manuscript to improve word choice, sentence structure, and grammar.  

Thank you for your comment. We have revised the English and adjusted the gramatical errors. 

Please describe the abbreviations when you use them for the first time

Revised. 

In the discussion part you need to compare with other studies from your country or surrounding regions. 

In the discussion, we compared our results to a similar study in Spain.
We mentioned as well important references of the topic in Italy published by Falcone, Tumbarello and Gianella.

I  think that you need three short clear conclusions! Make conclusion which can be used in current clinical use!

Thank you for the comment, we had added a conclusion focusing on 1. the role of source control, 2. the importance of appropriate timing of the antibiotic over the choice of mono or combination therapy, 3. the activity of novel antibiotics like ceftazidime-avibactam and current guidelines. 

Table 5 is hard to read! maximize writing!

Maximed.

Recheck  the references!

Revised. Thank you,

We hope that we had addressed the comments well.

Best regards, 

Round 2

Reviewer 1 Report

The authors modified the study as recommended.

According to the author guide, software cited need Citation reference. (Should be added)

Minor style and spell check are required.

Author Response

Dear Reviewer, 

Thank you for your positive comments.
We have added the citation of the statistical software. 

Best regards,